

# How Marine Emissions of Bromoform Impact the Remote Atmosphere

Yue Jia[1], Susann Tegtmeier[1], Elliot Atlas[2], Birgit Quack[1]

[1]GEOMAR Helmholtz Centre for Ocean Research Kiel, Kiel, Germany

[2]University of Miami, 4600 Rickenbacker Causeway, Miami, USA

*correspondence to*: Yue Jia (yjia@geomar.de)


**Abstract**
Oceanic emissions of very short lived halocarbons (VSLH), such as $CHBr_3$, are important for the halogen
budget of the atmosphere. It is an open question how localized elevated emissions in coastal and
upwelling regions and low background emissions, typically found over the open ocean, impact the
atmospheric VSLH distribution. In this study, we use the Lagrangian dispersion model FLEXPART to
simulate atmospheric $CHBr_3$ resulting from uniform background emissions, on the one hand, and from
elevated emissions observed during three tropical cruise campaigns, on the other hand.
The simulations demonstrate that the atmospheric $CHBr_3$ distributions due to uniform background
emissions are highly variable with accumulations taking place in regions of low wind speed. This relation
holds on regional and global scales demonstrating the importance of the atmospheric transport for the
distribution of short-lived trace gases with lifetimes in the range of days to weeks.
The impact of localized elevated emissions, measured during three research cruises, on the atmospheric
$CHBr_3$ distribution varies significantly from campaign to campaign. The estimated impact depends on
the strength of the emissions and the meteorological conditions. In the open waters of the western Pacific
and Indian Ocean, localized elevated emissions only slightly increases the background concentrations of
atmospheric $CHBr_3$, even when 1° wide source regions along the cruise tracks are assumed. Near the
coast, elevated emissions, including hotspots up to 100 times larger than the uniform background
emissions, can be strong enough to be distinguished from the atmospheric background. However, it is
not necessarily the highest hotspot emission that produces the largest enhancement, since the tug-of-war
between fast advective transport and local accumulation at the time of emission is also important.
Our analyses contribute to a better understanding and prediction of the timing and regional characteristics
of tropospheric $CHBr_3$ distribution. Significantly, our results demonstrate that transport variations of the
atmosphere itself are sufficient to produce highly variable VSLH distributions, and elevated VSLH in
the atmosphere do not always reflect a strong localized source. Localized elevated emissions can be
obliterated by the highly variable atmospheric background, even if they are orders of magnitude larger
than the average open ocean emissions.



## 1. Introduction

Very short lived halocarbons (VSLHs) with atmospheric lifetimes shorter than 6 months have natural oceanic sources which are dominated by brominated and iodinated compounds (Carpenter and Liss, 2000; Quack et al., 2004; Law et al., 2006). VSLHs have drawn considerable interest due to their contribution to stratospheric ozone depletion and tropospheric chemistry (Solomon et al., 1994; Dvortsov et al., 1999; Salawitch et al., 2005; Feng et al., 2007; Tegtmeier et al., 2015; Hossaini et al., 2015). In this work, we focus on the VSLH bromoform ($CHBr_3$), since most organic oceanic bromine is released into the atmosphere in this form.

$CHBr_3$ concentrations measured in ocean waters are characterized by large spatial variability with elevated abundances in phytoplankton blooms (Baker et al., 2000, Liu et al., 2013) and equatorial and upwelling regions due to biological sources (Carpenter et al., 2009; Quack and Wallace, 2003; Quack et al., 2007; Fuhlbrügge et al., 2016). The open ocean generally shows quite homogeneous, low $CHBr_3$ concentrations, compared to higher concentrations and strong gradients found in coastal and shelf areas (Quack and Wallace, 2003), At the coast, high oceanic concentrations are related to macro algae (Klick and Abrahamsson, 1992) and anthropogenic sources (Boudjellaba et al., 2016) such as power plants (Yang, 2001) and desalination facilities (Agus et al., 2009).

Due to sparse measurements and limited process understanding, existing estimates of global air-sea flux distributions of $CHBr_3$ and other VSLHs are subject to large uncertainties (e.g. Warwick et al., 2006; Palmer and Reason, 2009; Liang et al., 2010; Ordóñez et al., 2012; Stemmler et al., 2013; Ziska et al., 2013; Carpenter et al., 2014). Open ocean background emissions of $CHBr_3$ are modeled to be around 100 pmol m$^{-2}$ hr$^{-1}$ (Ziska et al., 2013), consistent with simultaneous in-situ measurements of air and water concentrations (Butler et al., 2007; Liu et al., 2013; Fiehn et al., 2017). The spatial and temporal distribution of elevated emissions in coastal and upwelling regions is currently based on very limited observations. Campaigns in these regions suggest that emissions generally increase near coastlines, and that sporadic peak emissions with extremely high values can be found (e.g. Butler et al., 2007; Liu et al., 2013; Fuhlbrügge et al., 2016; Fiehn et al., 2017). Analysis of the measurements suggests that such peak emissions are often of limited spatial extent and cover not more than a distance of 50-100 km along the cruise track. We will use the term 'elevated emissions' when describing emissions that are on average up to a factor of 10 larger than the background and 'hotspot emissions' for sporadic emissions up to a factor of 100 larger than the background.

There are two main approaches to derive the magnitude of VSLH emissions, i.e. "bottom-up" approach (e.g. Quack and Wallace, 2003; Carpenter and Liss, 2000; Butler et al., 2007; Ziska et al., 2013) and "top-down" approach (e.g. Warwick et al., 2006; Liang et al., 2010; Ordóñez et al., 2012). For the "bottom-up" method, measured surface sea water concentrations of VSLHs at the "bottom" (surface) are extrapolated to estimate global emissions. For the "top-down" method, the emissions of VSLHs are constrained by the measured abundances at the "top" (atmosphere) so that model simulations based on



the constrained global emission estimates reproduce the observed atmospheric concentrations. These two
approaches yield different estimates of the global VSLHs emissions, with the recent "top-down"
approaches resulting in generally higher emissions than the recent "bottom-up" approaches.
In the tropical ocean waters of the Atlantic, the western Pacific and Indian Ocean, the existence of
localized elevated $CHBr_3$ emissions and hotspots has been confirmed (Butler, et al, 2007; Liu et al., 2013;
Krüger and Quack, 2013; Quack and Krüger, 2013; Fiehn et al., 2017). At the same time, these
convectively active regions offer an efficient pathway for the vertical transport of short-lived oceanic
compounds from the boundary layer to the stratosphere (e.g. Aschmann et al., 2009; Hossaini et al., 2012;
Tegtmeier et al., 2012, 2013; Marandino et al., 2013; Liang et al., 2014). Moreover, the Asian monsoon
has been recognized as an efficient transport pathway for short-lived pollutants and VSLHs (Randel et
al., 2010; Hossaini et al., 2016; Fiehn et al., 2017). Given that elevated oceanic $CHBr_3$ emissions are
expected to occur in the same regions as strong convection, it is of interest to analyze how these elevated
emissions impact $CHBr_3$ in the atmospheric boundary layer, which feeds into the upward transport.
Measurements of $CHBr_3$ abundance in the atmospheric boundary layer show large spatial variability (e.g.,
Quack and Wallace, 2003; Montzka and Reimann, 2011; Lennartz et al., 2017). A compilation of
available measurements by Ziska et al. (2013) suggests similar $CHBr_3$ distribution patterns in the
atmospheric boundary layer as in the surface ocean, with higher mixing ratios in the equatorial, coastal
and upwelling regions. However, given the sparse data base and the uncertainties in the spatial and
temporal extent of oceanic emissions, the detailed distribution of boundary layer $CHBr_3$, cannot be well
constrained (e.g., Hepach et al., 2014; Fuhlbrügge, et al., 2013). On the one hand, the spatial and temporal
extent of elevated localized emissions is usually unknown, leading to large uncertainties when estimating
their overall magnitudes. On the other hand, the influence of meteorological conditions, distinctive
transport patterns and variations of atmospheric sinks, such as the background OH field (e.g. Rex et al.,
2014), can be expected to modulate the effect of elevated oceanic sources. Therefore, it is still an open
question of the magnitude of elevated and hotspot emissions on the local atmospheric $CHBr_3$ distribution.
Such knowledge is relevant to understand the importance of localized elevated emissions for atmospheric
abundances and to interpret existing atmospheric measurements with respect to potential sources and
driving factors.
In this study, we use observational data from three tropical research cruises, one in the Indian Ocean
(OASIS) and two in the western Pacific (TransBrom and SHIVA). We use the Lagrangian particle
dispersion model FLEXPART to investigate the transport and atmospheric distribution of VSLHs.
Taking bromoform ($CHBr_3$) as example, we compare the atmospheric signals estimated from the elevated
and hotspot emissions measured during the ship campaigns to the distribution derived from only uniform
background emissions. The campaigns and the FLEXPART model are introduced in Sect. 2. In Section
3, we discuss the distributions and variability of atmospheric $CHBr_3$ based on uniform background
emissions. We present the observed hotspots of $CHBr_3$ emissions in Section 4.1, and compare the



simulated atmospheric mixing ratios resulting from elevated emissions during three campaigns with the
background values (Section 4.2). Conclusions are given in Section 5.

**2. Data and Methods**

**2.1 Background and in-situ CHBr$_3$ emissions**

In this study, we distinguish between open ocean background and in-situ CHBr$_3$ emissions. Open ocean
emissions are deduced to be around 100 pmol h$^{-1}$ m$^{-2}$ based on global bottom up scenarios (Quack and
Wallace, 2003, Ziska et al., 2013). While emissions for individual regions and seasons can be higher or
lower than this, including negative fluxes going from the atmosphere into the ocean, 100 pmol h$^{-1}$ m$^{-2}$
represents the typical mean value averaged over all oceanic basins between 60°S and 60°N. The
background open ocean emissions exclude by design emissions from coastal, shelf and upwelling regions.

In-situ oceanic emissions of CHBr$_3$ have been calculated from the observational data collected during
three tropical ship campaigns. The two campaigns TransBrom (October 11th-23rd, 2009, Krüger and
Quack, 2013) and SHIVA (November 15th-28th, 2011, Quack and Krüger, 2013) took place in the
western Pacific, while the OASIS campaign (July 11th-August 6th, 2014, Fiehn et al., 2017) was
conducted in the western Indian Ocean. During each campaign, surface air and water samples were
collected simultaneously at regular intervals (every 3 to 6 hours). The emissions were calculated from
these co-located data and the instantaneous wind speed (Ziska et al., 2013, Fuhlbrügge et al., 2016, Fiehn
et al., 2017). The detailed cruise track and the magnitude of the oceanic CHBr$_3$ emissions of each
campaign is given in Fig. 1. The in-situ emissions include both open-ocean emissions and elevated
emissions from coastal, shelf and upwelling regions with our evaluations focusing on the regions of
elevated emission.

**2.2 Modeling**

For the simulations of the atmospheric distribution and transport of CHBr$_3$, we used the Lagrangian
particle dispersion model, FLEXPART (Stohl et al., 2005), which has been validated by previous
comparisons with measurements (Stohl et al., 1998; Stohl and Trickl, 1999). Lagrangian particle models
such as FLEXPART compute trajectories of a large number of so-called particles, presenting
infinitesimally small air parcels, to describe the transport, diffusion and chemical decay of tracers in the
atmosphere. The model includes turbulence in the boundary layer and free troposphere (Stohl and
Thomson, 1999) and a moist convection scheme (Forster et al., 2007) following the parameterization by
Emanuel and Živković-Rothman (1999). The representation of convection in FLEXPART simulations
has been validated with tracer experiments and $^{222}$Rn measurements (Forster et al., 2007). Chemical or
radioactive decay of the transported tracer is accounted for by reducing the particle mass according to a





prescribed lifetime of the tracer. Alternatively, the loss processes can be prescribed via OH reaction
based on a monthly averaged 3 dimensional OH-field. In this study, we employ FLEXPART version
10.0, which is driven by 3-hourly meteorological fields from ECMWF (European Centre for Medium-
Range Weather Forecasts) reanalysis product ERA-Interim (Dee et al., 2011) with a horizontal resolution
of 1º x 1º and 61 vertical model levels.
We performed two kinds of simulations based on the different emission scenarios. The first one used the
uniform global background emission, and the second one used in situ emissions observed during
individual ship campaigns. Chemical decay of $CHBr_3$ was simulated by prescribing a lifetime of 17 days
during all runs (Montzka and Reimann, 2011). For the background runs, a uniform air-sea flux of 100
pmol $h^{-1}$ $m^{-2}$ is prescribed over all ocean surface area between 60°S and 60°N. Three runs are conducted
covering the time period of the campaigns with a 1-month spin-up period in each case to reach a stable
background concentration in the atmosphere.
For the in-situ emissions of each campaign, simulations are based on the calculated $CHBr_3$ air-sea flux
(see Fig. 7, detailed description in Sec 4), which is released along the cruise track. The periods of these
campaign simulations are the same as the corresponding background simulations with emissions over
the whole time period. For each observational data point, an emission grid cell centered on the
measurement location is created. These grid cells are designed to be adjacent along the cruise track and,
based on the density of the measurements, are about $0.1 – 2.0°$ wide in cruise track direction. The grid
cells are chosen to be of a fixed width (0.5° or 1°) in the other direction and thus add up to the narrow
band of 0.5° or 1° width centered along the cruise track (Fig. 1). Our design of the emission grid cells
assumes that the elevated emissions can extend over a distance of 0.5°-1°. This choice has been motivated
by the spatial variability of the measurements along the cruise track (see also section 4.1 and Fig. 7).
Elevated emissions larger than 1000 pmol $h^{-1}$ $m^{-2}$ are found at 77 different locations along the three cruise
tracks examined in this paper. Out of the 77 measurements, only 11 correspond to singular locations with
no adjacent high emissions at the neighboring points. The other 66 measurements cluster together at 18
different locations with at least two adjacent observational points showing emissions larger than 1000
pmol $h^{-1}$ $m^{-2}$. We define the length of such a location of elevated emissions as the distance between the
first and last data point with an air-sea flux exceeding 1000 pmol $h^{-1}$ $m^{-2}$. Most of the 18 locations extent
over a distance larger than 0.5° (13 out of 18) and nearly half are larger than 1° (8 out of 18) supporting
our choice of the width of the emissions grid cells. Note that at the same time, the spatial extent of the
hotspots is comparable to the wind field resolution that drive our trajectory simulations. The amount of
$CHBr_3$ released from each grid cell is determined by the observational air-sea flux of the corresponding
data point and scales with the width of the narrow emission band described above. The specified $CHBr_3$
emission from each cell is kept constant for the duration of the model run and distributed over a fixed
number of trajectories. In order to capture the small scale processes (e.g. convection), the large number
of 2000, and 20000 trajectories are chosen to be released from each emission grid cell of background run
and in-situ run, respectively. Output data in form of $CHBr_3$ volume mixing ratios available at a user-
defined grid, is retrieved at a horizontal resolution of 1º x 1º and 0.5º x 0.5º for background runs and in-





situ runs, respectively, at every 100 m from 100 m to 1 km, and every 1 km from 1 km to 20 km every 3
hours.

**3. Atmospheric CHBr$_3$ based on open ocean background emissions**

The atmospheric CHBr$_3$ mixing ratios diagnosed from the uniform background emissions (referred to as
CHBr$_3$ background mixing ratios hereinafter) vary significantly from campaign to campaign and also
within each campaign region. Figures 2 to 4 present several snapshots of the CHBr$_3$ background mixing
ratios and the simultaneous wind fields from ERA-Interim reanalysis for the three campaigns. For
TransBrom (Fig. 2), CHBr$_3$ accumulates south of 15º N with a maximum near the equator, where the
wind is weak. In the northern Pacific, which was dominated by an anticyclone centered around 165°E,
30°N, the background values are much lower. On the 10[th] of October 2009, two bands of extremely low
wind fields exist, one directly south of the equator and one tilting from 15°N to 5°N, which both coincide
with the highest CHBr$_3$ abundances. On the 20[th] of October, these two bands collided into one with
lowest winds centered around 165°E, where we again find very high values of CHBr$_3$ of up to 0.8 ppt.
For both case studies, highest values are found in the region of the lowest wind speeds or slightly shifted
towards the region of strongest wind shear. Regions of high wind speeds, such as the northern Pacific
anticyclone, on the other hand do not allow for accumulations and are characterized by very low CHBr$_3$.
For the SHIVA case (Fig. 3), the background CHBr$_3$ accumulates in a narrow region near Indonesia,
with corresponding wind fields smaller than 3 m/s. North of Indonesia, the strong easterly trade winds
generally above 10 m/s prevent the accumulation of higher background values within the region. Again,
the two case studies illustrate how changes of the wind patterns within a few days drive changes of the
background CHBr$_3$ distribution. Another particular example is the northward extension of the low
equatorial winds around 90°E on 16[th] November 2011, which leads to higher CHBr$_3$ north of the equator
up to 15°N.
For OASIS (Fig. 4), the wind speed is higher than in the other two regions and these strong
southeast/southwest trade winds associated with the Asian monsoon extend over most of the Indian
Ocean. Consistent with the stronger winds, the background values for the OASIS case are significantly
lower than for the other two cases, although they also show accumulations in certain regions. These
accumulations appear partially in regions of low wind speeds (e.g., near the equator between 70°E and
90°E on 17[th] of July) or in adjacent regions of high wind shear (e.g., north of the equator between 70°E
and 90°E for both case studies). For the latter case, the CHBr$_3$ accumulation also extends into the region
of high wind speeds, which is different from the distribution found for the TransBrom and SHIVA
regions. This difference occurs because the east coast of the Indian Subcontinent offshore is a region
with wind convergence (not shown), which tends to accumulate air masses therein.
Given that the accumulation of CHBr$_3$ background mixing ratios follows in most cases the wind field
patterns on a regional scale, we hypothesize that the same relationship holds on a global scale. The global





distributions of atmospheric CHBr$_3$ based on background emissions and wind fields averaged over the
time periods of the SHIVA and OASIS cruise are presented in Fig. 5 and 6, respectively. We omit the
time period of the TransBrom case, since the background CHBr$_3$ distribution diagnosed for this period
is very similar to background found for the SHIVA period. The global CHBr$_3$ background mixing ratios
(Fig. 5a, and 6a) display a very heterogeneous distribution in spite of the uniform background emission
used for the simulations. Accumulations of CHBr$_3$ are again generally located in the regions of low wind
speeds. For the SHIVA period (November 2011), particularly high CHBr$_3$ background values of 0.3 to
0.4 ppt are found along the equator over the Maritime continent, West Pacific, Indian Ocean and at the
West coast of Africa, all of which are characterized by particularly low winds. In the Northern and
Southeast Pacific, the wind speed is generally higher, and the corresponding CHBr$_3$ values of less than
0.15 ppt are much lower than in the tropical region. For the OASIS period (July/August 2014), the global
CHBr$_3$ distribution is mostly reversed compared to the SHIVA period and high winds over the Indian
Ocean and Maritime continent lead to low CHBr$_3$ abundance in this region. The North Pacific on the
other hand, with low wind speeds is now a region of intense accumulation leading to 0.3-0.4 ppt of CHBr$_3$.
The tropical West Pacific is the only region that experiences relatively low winds during both seasons,
and constantly shows high CHBr$_3$ for the SHIVA and OASIS time periods.
The variations of the background CHBr$_3$ distribution can be generally explained by the seasonal
variations of the global wind field. The North Pacific and Northern Indian Ocean are dominated by the
East Asia Monsoon and the Monsoon of South Asia, respectively. The East Asia Monsoon is
characterized by strong northwesterly flow in boreal winter and weak southeasterly flow in boreal
summer due to the reverse of the thermal gradient between land and ocean (Webster, 1987; Ding and
Chan, 2005). Therefore, the accumulations of CHBr$_3$ in the North Pacific occurs during the boreal
summer months, rather than during boreal autumn/ early winter (TransBrom time period). The Monsoon
of South Asia, on the other hand, is characterized by weak northeasterly winds in boreal winter and
strong southwesterly winds in boreal summer (Webster, 1987; Webster et al., 1998). Thus background
CHBr$_3$ accumulation over the Northern Indian Ocean occurs mostly during boreal winter, while during
boreal summer (OASIS time period) a low CHBr$_3$ background can be expected. Because of the light
winds of the Inter Tropical Convergence Zone (ITCZ), a belt of relatively high CHBr$_3$ abundance exists
along the equator in the Northern Hemisphere, especially in the tropical Pacific and Atlantic. Strong
convection in the ITCZ enhances vertical transport of CHBr$_3$ out of the boundary layer, but overall the
CHBr$_3$ distribution is dominated by the horizontal wind fields and accompanying transport patterns. Due
to the more complex land-sea thermal difference, the seasonal variation of ITCZ in the West Pacific is
more significant than in the East Pacific (Waliser and Jiang, 2014). The relatively high accumulations of
CHBr$_3$ in the tropical East Pacific are confined to a narrow region near the equator for both seasons. As
for the tropical West Pacific, during boreal winter the ITCZ covers almost the whole Southeast Asia and
the high CHBr$_3$ abundances during SHIVA appear along the east coast of Malaysia. During boreal
summer, the ITCZ shifts northward and the high CHBr$_3$ abundances retreat northwestward.



We assume constant CHBr$_3$ open ocean emissions of 100 pmol h$^{-1}$ m$^{-2}$ for our simulations in order to
isolate the impact on the atmospheric CHBr$_3$ distribution of atmospheric transport patterns versus the
impact of varying emission fields. In particular, variations of the wind fields will impact the ocean air-
sea flux, and emissions larger than 100 pmol h$^{-1}$ m$^{-2}$ might occur in regions of higher winds with little
CHBr$_3$ accumulation. Such variations can change the background CHBr$_3$ distribution and may allow for
increased mixing ratios in regions of strong winds. In addition to the wind speed, variations in the
atmospheric and, more importantly, the oceanic CHBr$_3$ concentrations can impact the emission strength
which can further change the complex atmospheric CHBr$_3$ distribution.

**4. Atmospheric CHBr$_3$ based on hotspot emissions**

Given the high variability of the atmospheric CHBr$_3$ background mixing ratios, resulting from
atmospheric transport processes (Section 3), it is of interest to analyze if and how much oceanic hotspot
emissions might impact this background distribution. In this section, we will use observational data to
discuss if oceanic hotspot emissions occur at the same time and location as peak atmospheric mixing
ratios or if the two quantities are rather uncorrelated. Furthermore, we will use FLEXPART simulations
to compare CHBr$_3$ mixing ratios that result from background emissions to the increased CHBr$_3$ mixing
ratios that result from localized hotspot emissions.

**4.1 Observed hotspot emission**

Oceanic CHBr$_3$ emissions, atmospheric CHBr$_3$ mixing ratios and the observed local surface wind speeds
are given in Fig. 7 for all three campaigns. The oceanic emissions of CHBr$_3$ vary substantially from
campaign to campaign with mean values of 261 pmol h$^{-1}$ m$^{-2}$ (TransBrom), 1228 pmol h$^{-1}$ m$^{-2}$ (SHIVA),
and 912 pmol h$^{-1}$ m$^{-2}$ (OASIS) with standard deviations of 600 pmol h$^{-1}$ m$^{-2}$ (TransBrom), 1460 pmol h$^{-1}$
m$^{-2}$ (SHIVA) and 1159 pmol h$^{-1}$ m$^{-2}$ (OASIS), respectively. All three campaigns show periods with
background emissions around 100 pmol h$^{-1}$ m$^{-2}$ and periods with localized elevated and hotspot emissions.
For TransBrom, the first two thirds of the campaign show negative (air-to-sea) or very low background
CHBr$_3$ fluxes, while the last third was close to western Pacific islands and is characterized by overall
elevated emissions with sporadic hotspots of up to 4000 pmol h$^{-1}$ m$^{-2}$. The SHIVA cruise track, on the
other hand, was mostly along the coastline, and low background emissions occur only for brief periods.
Most locations showed elevated emissions and hotspots occurred regularly. The OASIS cruise track
alternated between open ocean, upwelling and coastal areas, resulting in a large fluctuation between low
background and localized elevated emissions. Hotspot emissions during this campaign are largest,
reaching values of over 6000 pmol h$^{-1}$ m$^{-2}$.
According to the flux parameterization applied here, the air-sea flux is determined mostly by the surface
wind speed and the ocean-atmosphere concentration gradient. Highest emissions are expected to occur





during periods of high wind speed and large concentration gradients. The wind speed dominates the
fluxes for some regions, but not for entire campaigns. During the beginning of the TransBrom campaign
(Fig. 7a), the wind speed peaks at over 15m/sec while the corresponding $CHBr_3$ air-sea flux is low.
Higher wind speeds co-occur with high air-sea fluxes at the end of the campaign. For SHIVA (Fig. 7b)
and OASIS (Fig. 7c), the relation between wind speed and $CHBr_3$ emissions is more easily discernable.
All three campaigns demonstrate that high fluxes do not always lead to local high $CHBr_3$ mixing ratios
in the surface atmosphere. For example, several hotspots with oceanic emissions over 4000 pmol $m^{-2}$ $hr^{-}$
$^1$ are found during OASIS, however, corresponding atmospheric mixing ratios are relatively low (~ 2
ppt). Vice versa, the highest atmospheric mixing ratios found during OASIS do not coincide with high
fluxes, except for the last part of the campaign. These discrepancies suggest that the local atmospheric
mixing ratios are driven by a complex interplay of source and loss processes driving the atmospheric
mixing ratios of short-lived compounds. A relatively clear connection between elevated oceanic
emissions and surface mixing ratios only occurs during the SHIVA campaign and during the last part of
the TransBrom campaign (Fig 7a and b).
As the campaign data shows a relation between emissions and atmospheric mixing ratios only for some
regions, the question arises how much of the atmospheric variability of short-lived compounds such as
$CHBr_3$ is impacted by the emission strengths. In order to quantify the relative contributions of elevated
emissions, comparisons between the computed mixing ratios over hotspots and the background
abundances in the atmosphere are required. In the subsequent section, we present such comparisons based
on the model results.

**4.2 Comparison of $CHBr_3$ from background and hotspot emissions**

In this section, we will compare the concentrations of $CHBr_3$ due to background and localized elevated
emissions as simulated by FLEXPART. First, atmospheric $CHBr_3$ during all three campaigns is
calculated based on the uniform background emission of 100 pmol $h^{-1}$ $m^{-2}$. Second, atmospheric $CHBr_3$
resulting from strong localized emissions is simulated for the three case studies given by the campaigns.
Atmospheric $CHBr_3$ at different altitudes is simulated by FLEXPART, which is driven by the
meteorological data from ECMWF. The signatures of dynamical processes such as wind regimes,
weather phenomena (e.g., typhoons) and convection are captured by the model simulation and can be
detected in the $CHBr_3$ distribution (Fig. 8). For example, during the TransBrom campaign, the cruise
encountered several tropical storms in the western Pacific, one of which (Lupit, around October 14th,
2009) developed into a super typhoon within several days (Krüger and Quack, 2013). As shown in Fig.
8, an elevated $CHBr_3$ accumulation representing the structure of typhoon Lupit is clearly visible in the
background distribution of $CHBr_3$ at 500 m (Fig. 8d) altitude in the northern part of the western Pacific.
This structure is still clear at 5 km altitude (Fig. 8a), although with a weaker magnitude. For $CHBr_3$
emitted from localized elevated sources (Figures 8b, c, e, and f), such large scale structures are not



discernible due to the small spatial extent of the 0.5° or 1° emission cells and thus the limited amount of
overall released $CHBr_3$. A clear blob of higher abundances of atmospheric $CHBr_3$ can be seen in the
southern part of the western Pacific near Indonesia resulting from one of the hotspot emissions observed
during TransBrom (Fig. 1b). However, the background $CHBr_3$ in this area is also high in this low-wind
area, and thus the atmospheric signal of the up to 20 times stronger hotspot emissions (Fig. 7a) is
detectable for neither the 0.5° nor the 1° wide emission cells when compared to the background. Note
that the modelled atmospheric mixing ratios from both sources, hotspot and background emissions, are
smaller than the mixing ratios observed along the cruise track (Figure 7) suggesting stronger nearby
emissions not covered in our scenarios and observations. The signature of the hotspot emissions remains
in the boundary layer and cannot be seen at 5 km altitude.
Fig. 9 shows the atmospheric $CHBr_3$ mixing ratios during the SHIVA campaign. For the SHIVA case,
the distribution patterns of the background $CHBr_3$ abundances at 500 m altitude also show a very strong
spatial variability, despite the uniform emissions. Highest $CHBr_3$ background mixing ratios around 120º
E near the equator of up to 0.5 ppt are smaller than background values found during TransBrom of up to
0.8 ppt. The atmospheric signal of the localized elevated emissions is much stronger than during
TransBrom due to stronger emissions on the one hand and smaller background mixing ratios on the other
hand. First, for the 0.5° wide emission grids, two highly localized, atmospheric $CHBr_3$ peaks appear
close to the coast line near the equator around 105º E with a maximum value around 0.4 ppt. These
signals occur in a spot where the background is very low (0.2 ppt). However, at the same time they are
smaller than the maximum background values of up to 0.5 ppt in nearby regions (Fig. 9d). If the width
of the emission grids is extended to 1°, the localized $CHBr_3$ peaks mentioned above grow into two distinct
blobs near the equator of up to 0.8 ppt. These maxima along the first half of the cruise track are apparently
larger than the regional background concentrations (Fig. 9f). Elevated emissions during the second half
of the campaign with several hotspot events, on the other hand, do not show such clear atmospheric
signals right above.
At 5 km altitude, the overall background values are slightly lower, but the maximum mixing ratios are
still comparable to altitudes below due to the intense convection in the general region of the SHIVA
campaign (Fuhlbrügge et al., 2016). For the regions of localized elevated emissions, the convection is
less effective and maximum mixing ratios at 5 km are about 50% smaller compared to the values in the
boundary layer. Krystofiak et al. (2018) calculated the fractions of convective-contributed trace gases
from boundary layer to the upper troposphere using airborne measurements during the SHIVA campaign
and reported an even smaller fraction of boundary layer $CHBr_3$ in the upper troposphere (about 15% due
to convection). Thus only the signal of the 1° wide emission cells can be detected at 5 km, while assuming
that the emissions cover a smaller region of 0.5° width will render their impact in the free troposphere
negligible.
Due to the dominant southwest monsoon over the Northern Indian Ocean in boreal summer, the resulting
atmospheric abundances of the OASIS case (Fig. 10) for both scenarios, background and localized



emissions, are much lower than for the other two campaigns. This is particularly surprising for the OASIS
hotspot emissions, which are in many cases larger than hotspot emissions during TransBrom or SHIVA.
In the open ocean, the atmospheric enhanced $CHBr_3$ mixing ratios resulting from the 0.5° (1°) wide
localized emission runs reach only 0.1 (0.2) ppt in a narrow belt near 60°E and are mostly smaller than
the background (around 0.15 ppt). An exception occurs near the coast of Madagascar, where both
background and hotspot emissions accumulate in the atmosphere. Maximum background values reach
up to 0.25 ppt and the hotspot signals peak with values of 0.3 ppt (0.5° wide emission cells) to 0.6 ppt
(1° wide emissions cells). These clear atmospheric signals of hotspot emissions are driven by the
enhanced coastal emissions near Madagascar. At 5 km altitude, atmospheric background values are very
low, and the hotspot contributions are close to zero.
In summary, the observed emissions during the three cruises were significantly higher than the
background of 100 pmol m$^{-2}$ hr$^{-1}$. Our results show that such strong oceanic sources are not necessarily
detectable in the atmosphere, where transport processes can sometimes mask the impact of oceanic
emissions on the atmospheric $CHBr_3$ distribution.

**5. Summary and Discussion**

In this study, we simulated atmospheric $CHBr_3$ abundances that result from uniform marine background
emissions compared to hotspot emissions using the Lagrangian dispersion model FLEXPART.
The simulations demonstrate that uniform background emissions from the ocean result in a highly
variable atmospheric $CHBr_3$ distribution with accumulations taking place in regions of low wind speed.
This relation holds on regional and global scales revealing atmospheric transport processes as important
drivers of the distribution of short-lived trace gases with lifetimes in the range of days to weeks. The
relation between atmospheric background and wind patterns described here will allow us to better predict
the seasonal and regional characteristics of the tropospheric $CHBr_3$ distribution. Such knowledge will
provide valuable information for analyzing and interpreting atmospheric data from ship and aircraft
campaigns. For example, our results illustrate that low atmospheric $CHBr_3$ abundances cannot
necessarily be used to draw conclusions about the oceanic source strength below.
Comparisons between atmospheric $CHBr_3$ resulting from background and peak emissions suggest that
the impact of localized elevated emission on the atmospheric $CHBr_3$ distribution depends on their relative
strength, on their location and on the time of emission. The "visibility" of elevated emissions in the
atmospheric $CHBr_3$ distribution varies significantly between three cruises in the West Pacific and Indian
Ocean. In the open ocean, signals of elevated emissions can hardly be distinguished from the background
$CHBr_3$ distribution even for elevated sources extending over 1° wide source regions along the cruise
tracks. Near the coast, however, signals of elevated emissions are often strong enough to be distinguished
from the background. In particular, some of the hotspot emissions up to 100 times larger than the
background can be detected in the atmosphere. However, individual cases show that it is not necessarily





the largest hotspot that gives a clear signal, but that the tug of war between fast advective transport and
local accumulation at the time of emission is also important.
The constant background emissions of 100 pmol $m^{-2}$ $hr^{-1}$ used in our study are based on a simplified
scenario and do not take coastal and upwelling maxima into account. Realistic oceanic emissions are
much more complex with large gradients in the above mentioned regions. Nevertheless, our results
demonstrate that atmospheric $CHBr_3$ signals, produced by localized elevated and even hotspot emissions,
orders of magnitudes larger than the average open ocean emissions, can be obliterated by the highly
variable atmospheric background. That is to say that transport variations of the atmosphere itself are
sufficient to produce high concentrations in certain regions and that high concentrations of VSLH in the
atmosphere do not always guarantee a strong local or regional source. For observational and modelling
studies of VSLS and other short-lived compounds, the impact of atmospheric transport patterns that are
identified here can be used for the interpretation of trace gas distributions and variability.




**Data availability**

The emission data of cruise campaigns are available at Pangaea (http://www.pangaea.de). FLEXPART
output can be inquired from the authors.

**Author contribution**

Y. Jia, S. Tegtmeier designed the model experiments. Y. Jia carried out the FLEXPART calculations and
produced the figures. Y. Jia and S. Tegtmeier wrote the manuscript with contributions from all co-authors.

**Competing interests**

The authors declare that they have no conflict of interest.

**Acknowledgements**

The authors would like to thank the European Centre for Medium-Range Weather Forecasts (ECMWF)
for the ERA-Interim reanalysis data and the FLEXPART development team for the Lagrangian particle
dispersion model used in this publication. The FLEXPART simulations were performed on resources
provided by the computing center at Christian–Albrechts–Universität in Kiel.



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


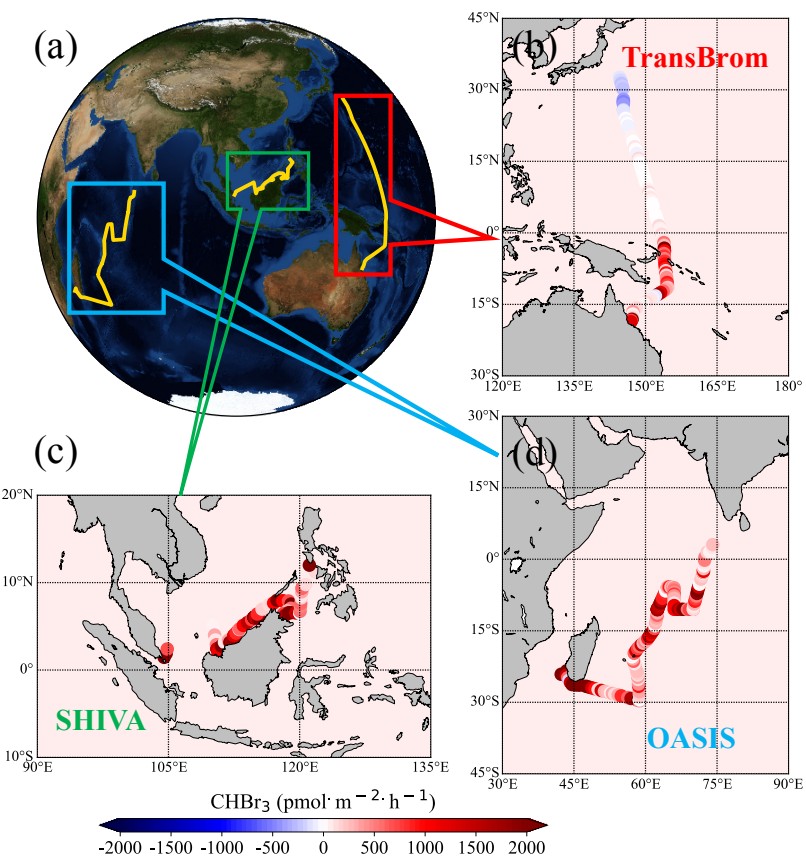


**Fig 1.** Cruise tracks of the three campaigns in the Indian Ocean and Western Pacific (a)
and $CHBr_3$ emissions (b, c, d) used in the model simulation. Global background
emissions (100 pmol m$^{-2}$ hr$^{-1}$) and observed emissions along the tracks of the three
research cruises TransBrom (b), SHIVA (c), and OASIS (d).









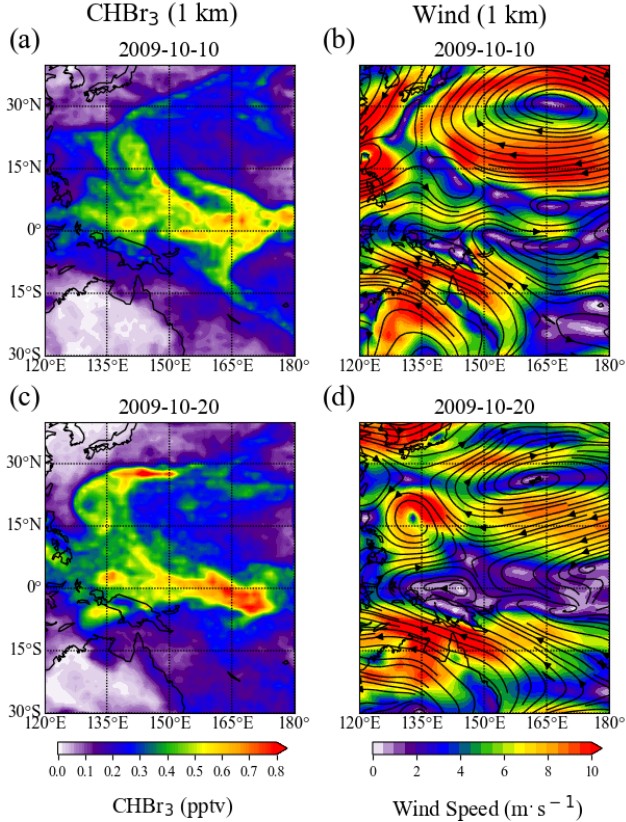



**Fig. 2.** Two snapshots of spatial distributions of atmospheric CHBr$_3$, derived from
uniform oceanic background emissions of 100 pmol m$^{-2}$ hr$^{-1}$ (a, c), and ERA-Interim
reanalysis wind fields (b, d) at 1 km altitude during TransBrom. The wind speed is
denoted by color shades and the directions are denoted by the stream lines.




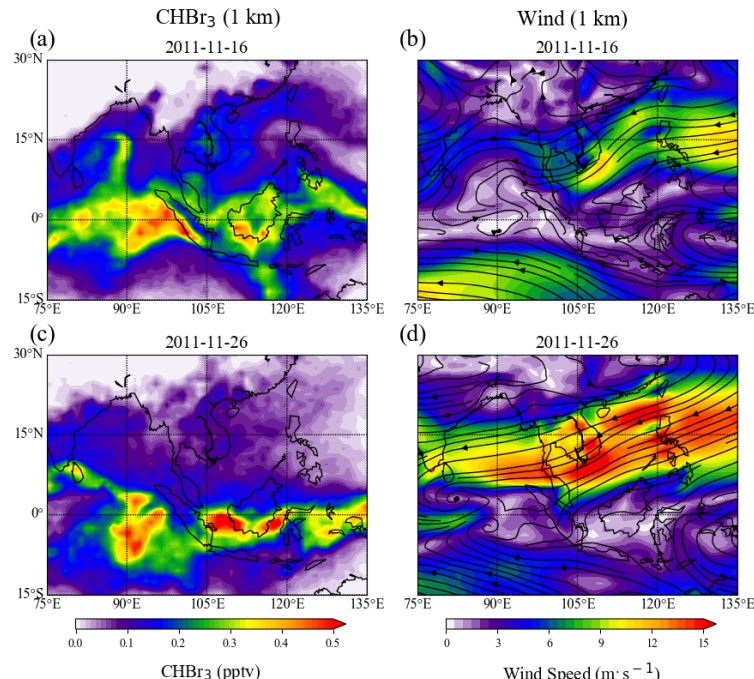


**Fig. 3.** Same as Fig. 2, but for SHIVA case.






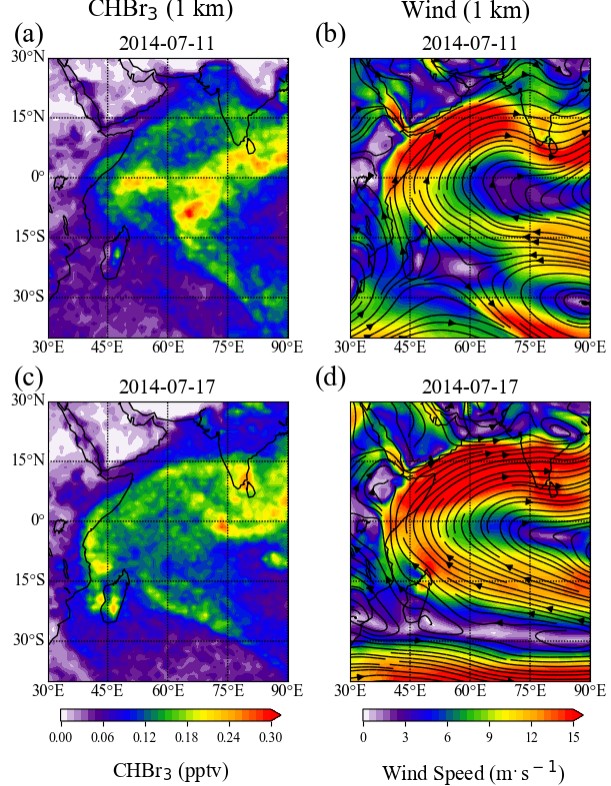


**Fig. 4.** Same as Fig. 2, but for OASIS case.





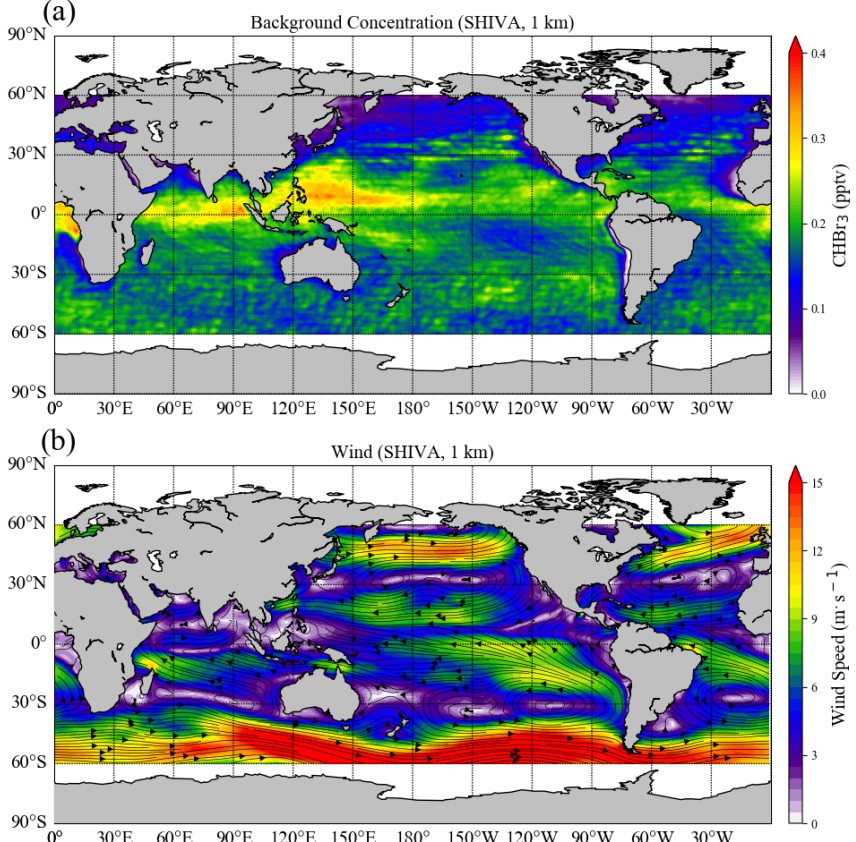

**Fig. 5.** Global distributions of CHBr$_3$ mixing ratios based on oceanic background emissions (a), and ERA-Interim reanalysis wind fields (b) averaged during the time period of the SHIVA cruise at 1 km. The wind speeds are denoted by color shades and the directions are denoted by the stream lines.





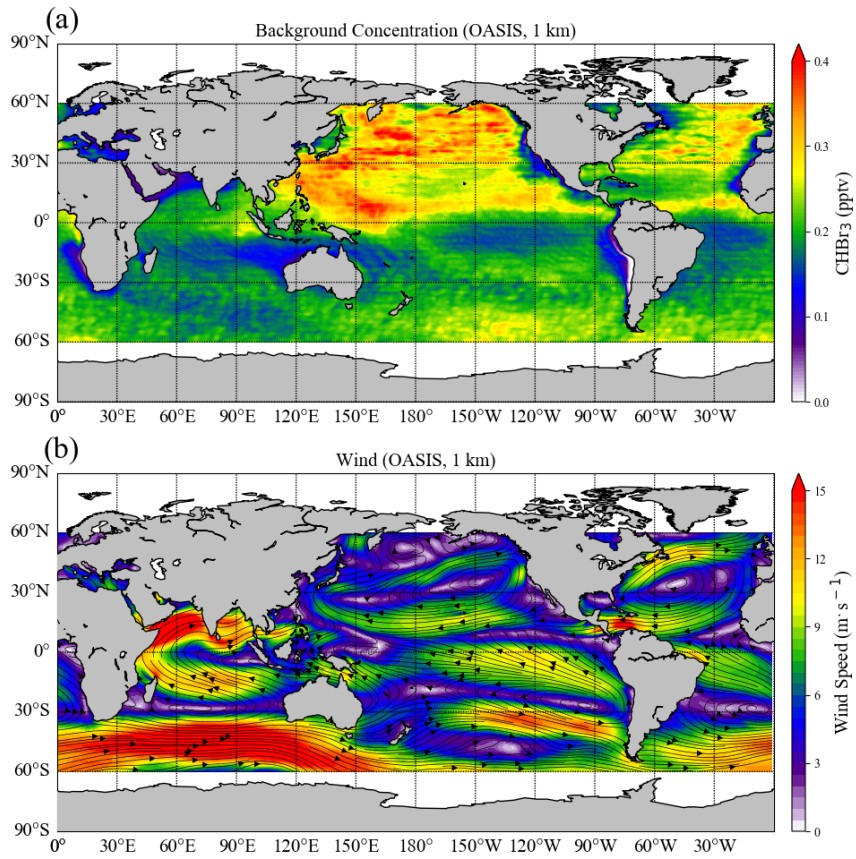


**Fig. 6.** Same as Fig. 5 but for OASIS case.




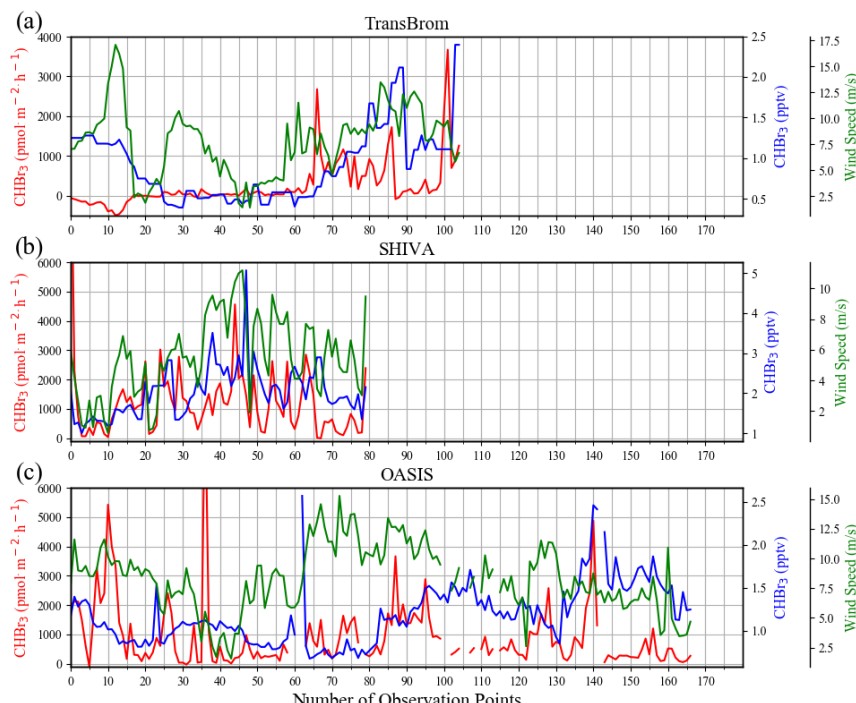



**Fig. 7.** Surface wind speeds (green), CHBr$_3$ air-sea flux (red), and atmospheric mixing
ratios of CHBr$_3$ near surface (blue) observed during TransBrom, SHIVA, and OASIS
campaigns.

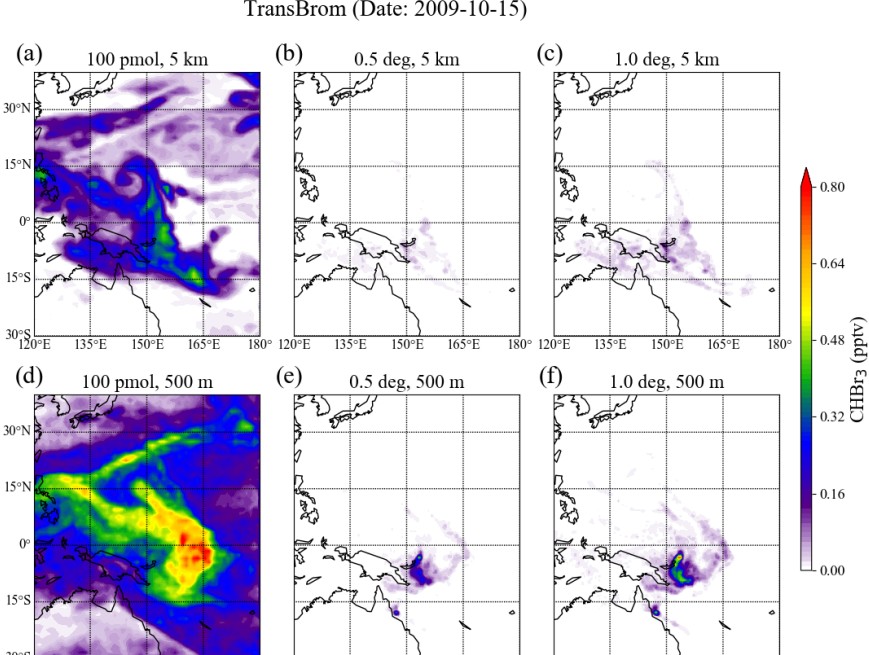


**Fig. 8.** Atmospheric CHBr$_3$ mixing ratios at different altitudes (500 m and 5 km)
simulated for the time period of the TransBrom campaign. Simulations are based on
background emissions (a, d), and elevated emissions observed during the campaign for
0.5° (b, e) and 1° (c, f) wide emission grid cells.



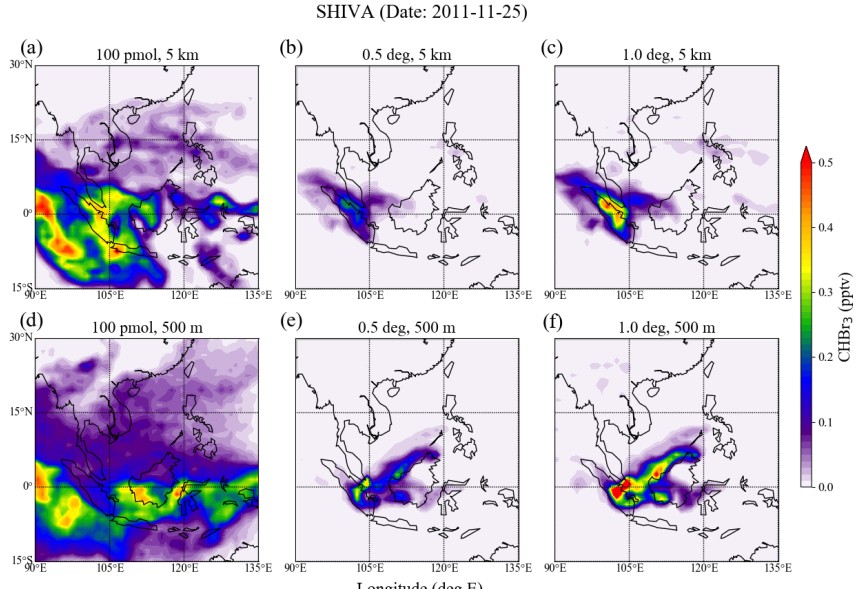


**Fig. 9.** Same as Fig. 8, but for the SHIVA campaign.



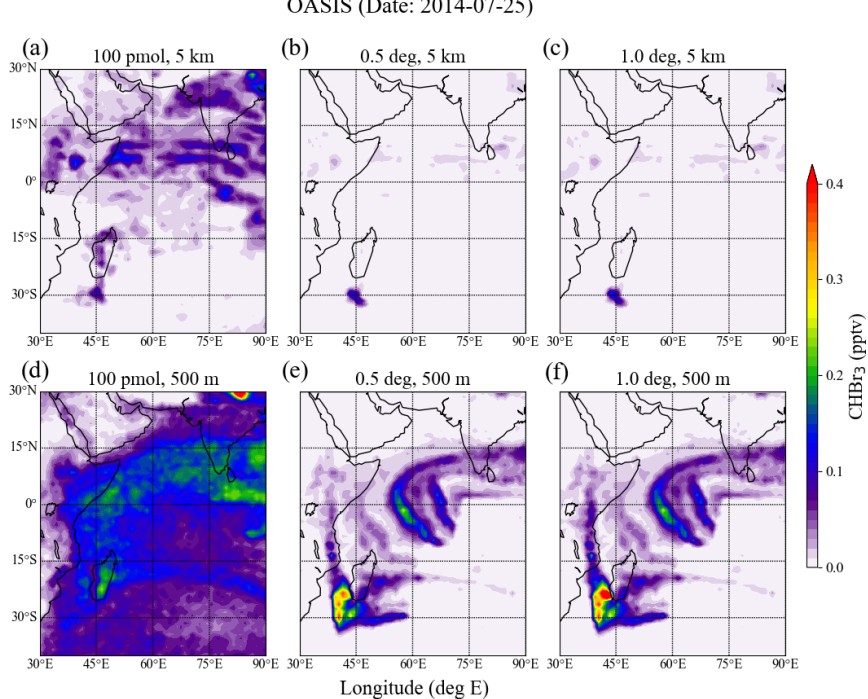


**Fig. 10.** Same as Fig. 8, but for the OASIS campaign.