# Peer review of "How Marine Emissions of Bromoform Impact on the"

_Atmospheric Chemistry and Physics, 2018_

## Referee Comment (RC1) · Anonymous Referee #1 · 14 Jan 2019

I have read the paper, entitled "How Marine Emissions of Bromoform Impact the Remote Atmosphere", by Yue Jia et al. and find it needs some revision before publication. The paper quantitatively underscores the interplay between emissions, loss or degradation, and transport in determining the concentrations of bromoform in the atmosphere. It is especially valuable because of its use of observations to test model results. It would be nice if the authors went further and used other oceanic and atmospheric observations that are available as well, but the paper as it stands is a good start on using this kind of approach to understand the behavior of CHBr3 in the atmosphere. My biggest concern is that the discussion bounces around too much, making it difficult to draw out the major points of the paper. If the authors can fix that and focus the paper on the major points, it should be good to go.

[Figure]

General Comments

The authors need to be sure they are clear when discussing models and observations. Certain parts of the paper have to be read a couple of times to be sure which is which, and we don't want the readers confusing simulations with reality. Also, be specific and consistent with terminology. "Boundary layer", for example, could replace "atmosphere" in several places in the paper. It's good to avoid using the word "concentration" to describe atmospheric amounts, which are better expressed as "mixing ratio" or "mole fraction". Overall, I think reorganizing the discussion around the major points they want to make would make the paper easier to understand and put less of a strain on the reader.

Specific comments

Title: Add "on" after "Impact"

Abstract: Delete first sentence Line 12: Insert "of CHBr3 and other VSLH's" after "emissions". Line 13: Insert "on" after "impact". Line 15: Insert "assumed" before "uniform"; delete "on the one hand". Line 16: Insert "consistent with those" before "observed". Emissions were not measured on those cruises, but calculated using a variety of assumptions. Delete "on the other hand". Line 17: Replace "due to" with "in the". Line 18: Insert "scenario" after emissions. Lines 19&20: Insert comma after "scales"; delete "the" before "atmospheric"; replace "the distributions of" with "distributing". Line 24: Insert "s" after "Ocean"; change "increases" to "increase". Lines 30&31: Delete sentence. It's too general, maybe not entirely true, and for others in the world to decide. Line 31: Delete "Significantly" and capitalize "Our". Line 32: Second "VSLH" should probably be "VSLHs" to denote plural.

Introduction:

Line 40: Replace "which" with "that". Line 49: Delete "quite". Line 60: Delete "very". Line 71: Insert "ocean" before "surface".

Data and Methods

Background

First paragraph, second sentence is redundant. This was already said in the introduction. I think the specificity belongs here in the methods, but the attribution (citations) for the two statements is different. The authors might consider using the attributions for the intro statement here and then deleting the sentence in the intro, thus keeping the intro general. Or they could omit it here if they prefer.

Line 133: Insert comma after "shelf".

Modeling

Lines 160-164 and elsewhere in the methods section: Try to keep to one tense. Use of present tense for something that has already happened or already been done makes for awkward reading, especially when mixed with past tense to describe the same. Prefer present tense for describing things that are already understood or "true" in the broader sense.

Atmospheric CHBr3 (background). . .

Although it contains some good points, I find this section meandering and confusing. I believe it would benefit from some reorganization. I can derive what the authors are getting at, but only after several readings. Because this section is about what was learned from the transport model with constant emissions, it might be an easier read if the authors could state simply what the model demonstrates, then point out how data from each cruise supports those points in the model or where they differ. (But I think that is for the discussion on "hot spots", or am I missing something?) For example, the statement on lines 239-240 summarizes a major point. It would be good to lead with that or another similar statement, then follow with a discussion about the features it demonstrates, and then go to the data if appropriate. Then go to another point in a similar fashion.

BTW, lines 260-262 are redundant for a second time (i.e., third mention), but if the authors want to reiterate that point, it should be at the beginning of the section, not in the middle.

Some specific points here that may or may not be helpful:

Line 191: I believe the authors mean "boundary layer mixing ratios" or "mixing ratios in the lower atmosphere" rather than "atmospheric" here? Lines 215 and 220: I think I know what the authors are trying to say, but it's not possible to accumulate the amount of a gas through transport and mixing, and these statements come across as though that's what they are saying. I think they want to state this more carefully, perhaps invoking how wind engages with the ocean surface while diluting what is emitted? Or is it simply about how the winds and mixing and loss redistribute CHBr3? Line 222: It's really not a big leap to hypothesize that winds reduce the mixing ratios in the boundary layer. I think the valuable story is about how the wind shear and degradation can influence the distributions. Lines 224-225: "time period" is redundant; "respectively" is almost always unnecessary. Lines 228-229: Again, are these actually accumulations or higher concentrations as the result of dispersal? The former requires emissions of some sort and I don't think that is what this section is about (i.e, constant emissions everywhere).

Atmospheric CHBr3 (hotspots)

Observed

Lines 289-90. Delete "on the other hand" Line 290: What is meant by "background" here? I would not use "background' in this paper in any sense other than defined in lines 191-92. Lines 297-298: How does windspeed dominate the flux? Do the authors mean it is dominating the variability in the flux? Lines 306-308: What happened to transport? It's not all just source and loss. And I'm curious, how is loss being handled? I presume it is variable in the atmosphere based on absorption bands etc.? Lines 339-342: Figure 7 is observations, not models. What is meant here? Lines 349-350:

[Figure]

Delete the "hand" phrases. Line 357: "Hand" again. Delete. Lines 381-384: This is an important statement. Don't lose it during the revision.

Summary

Just a thought. The summary could be trimmed and used as the abstract of the paper, as it focuses on "what we did, what we found, and what it means", which is all that is really needed in an abstract.

Line 392: Replace "revealing" with "demonstrating the role of"; delete "important". Line 401: Insert comma after "location". Line 416: Replace "produce" with "allow".

---

## Referee Comment (RC2) · Anonymous Referee #2 · 30 Mar 2019

The authors present an interesting model study on the spatial variation of atmospheric CHBr3 concentrations. Overall the paper is clearly written, and easy to read. While I agree their main conclusion that the spatial distribution is a result of the interplay of transport and emissions (and chemical depletion), I think a major revision is needed before it can be accepted for publication. My major concerns are:

1. There is no sufficient description on their transport model. The FLEXPART model has been widely used in various researches. But for CHBr3 simulation, the reader still wants to know the important details, including its description on the air-sea exchange, and on the oxidation scheme etc.

2. There is no comparison of their simulations with other models, or more importantly with observations. There are questions on how realistic their simulations are. For example, the model shows large areas with very low CHBr3 concentrations (<0.1 pptv), particularly at the 5km level (Figs.8 to 10). They appear inconsistent with aircraft measurements such as the recent CAST and CONTRAST campaigns.

3. More detailed analysis is needed. At some places, the paper tends to establish the correlation between the CHBr3 distribution and the (snapshot of) wind fields. Considering the lifetime of CHBr3, such correlations are often not obvious, as demonstration by their own results. Tagged simulations (for example, Butler et al., 2018) may help the reader understand the complexity, particularly for the emission 'hotspots'.

---

## Author Comment (AC1) · 29 May 2019

Dear Referee #1

We thank the referee for his/her valuable comments, which helped to improve the manuscript. We have reorganized our discussions to make the main point clearer and revised other comments accordingly. The changed parts in the manuscript are marked in red. The following is a point-by-point response.

**Anonymous Referee #1**

**I have read the paper, entitled "How Marine Emissions of Bromoform Impact the Remote Atmosphere", by Yue Jia et al. and find it needs some revision before publication.**
**The paper quantitatively underscores the interplay between emissions, loss or degradation, and transport in determining the concentrations of bromoform in the atmosphere.**
**It is especially valuable because of its use of observations to test model results. It would be nice if the authors went further and used other oceanic and atmospheric observations that are available as well, but the paper as it stands is a good start on using this kind of approach to understand the behavior of CHBr3 in the atmosphere.**

Answer: We added comparisons with aircraft campaigns of CAST and CONTRAST and other model studies in the discussion session.

**My biggest concern is that the discussion bounces around too much, making it difficult to draw out the major points of the paper. If the authors can fix that and focus the paper on the major points, it should be good to go.**
**General Comments**
**The authors need to be sure they are clear when discussing models and observations. Certain parts of the paper have to be read a couple of times to be sure which is which, and we don't want the readers confusing simulations with reality. Also, be specific and consistent with terminology. "Boundary layer", for example, could replace "atmosphere" in several places in the paper. It's good to avoid using the word "concentration" to describe atmospheric amounts, which are better expressed as "mixing ratio" or "mole fraction". Overall, I think reorganizing the discussion around the major points they want to make would make the paper easier to understand and put less of a strain on the reader.**
Answer: We revised our manuscript according to the suggestions, especially the discussion parts. See detailed responses below.

**Specific comments**
**Title: Add "on" after "Impact"**

Answer: Added

**Abstract: Delete first sentence Line 12: Insert "of CHBr3 and other VSLH's" after "emissions".**
**Line 13: Insert "on" after "impact". Line 15: Insert "assumed" before "uniform"; delete "on the one hand". Line 16: Insert "consistent with those" before "observed".**
Answer: Corrected as suggested.

**Emissions were not measured on those cruises, but calculated using a variety of assumptions.**
Answer: "observed" is replaced by "derived"

**Delete "on the other hand".**
Answer: Deleted.

**Line 17: Replace "due to" with "in the".**
**Line18: Insert "scenario" after emissions.**
**Lines 19&20: Insert comma after "scales"; delete "the" before "atmospheric"; replace "the distributions of" with "distributing".**
**Line 24: Insert"s" after "Ocean"; change "increases" to "increase".**
Answer: Revised as suggested.

**Lines 30&31: Delete sentence. It's too general, maybe not entirely true, and for others in the world to decide.**
**Line 31: Delete "Significantly" and capitalize "Our". Line 32: Second "VSLH" should probably be "VSLHs" to denote plural.**
Answer: The sentence of Lines30&31 is deleted and the other parts are revised as suggested.

**Introduction:**
**Line 40: Replace "which" with "that". Line 49: Delete "quite". Line 60: Delete "very".**
**Line 71: Insert "ocean" before "surface".**
Answer: Corrected as suggested.

**Data and Methods**
**Background**
**First paragraph, second sentence is redundant. This was already said in the introduction. I think the specificity belongs here in the methods, but the attribution (citations) for the two statements is different. The authors might consider using the attributions for the intro statement here and then deleting the sentence in the intro, thus keeping the intro general. Or they could omit it here if they prefer.**

Answer: We just keep the sentence in this part and the sentence in the intro section is deleted.

**Line 133: Insert comma after "shelf".**

Answer: Corrected.

**Modeling**
**Lines 160-164 and elsewhere in the methods section: Try to keep to one tense. Use of present tense for something that has already happened or already been done makes for awkward reading, especially when mixed with past tense to describe the same. Prefer**

**present tense for describing things that are already understood or "true" in the broader sense.**

Answer: We went through the method section and corrected the tenses as suggested.

**Atmospheric CHBr3 (background). . .**

**Although it contains some good points, I find this section meandering and confusing. I believe it would benefit from some reorganization. I can derive what the authors are getting at, but only after several readings. Because this section is about what was learned from the transport model with constant emissions, it might be an easier read if the authors could state simply what the model demonstrates, then point out how data from each cruise supports those points in the model or where they differ. (But I think that is for the discussion on "hot spots", or am I missing something?) For example, the statement on lines 239-240 summarizes a major point. It would be good to lead with that or another similar statement, then follow with a discussion about the features it demonstrates, and then go to the data if appropriate. Then go to another point in a similar fashion.**

Answer: We reorganized this part with the suggested structure. Each of the paragraph begins with a summarized statement and followed by detailed discussions.

The sentences at the beginning of the paragraphs in the manuscript are: "In this section, we show the impact of atmospheric transport patterns on the atmospheric $CHBr_3$ distribution, with the uniform background $CHBr_3$ emission simulations." in lines 204-205 "The relationship mentioned above also holds on a global scale" in line 254; and "In the above simulations, we assume a constant background emission in order to isolate the impact of transport and loss processes on the atmospheric $CHBr_3$ distribution." in lines 293-294.

**BTW, lines 260-262 are redundant for a second time (i.e., third mention), but if the authors want to reiterate that point, it should be at the beginning of the section, not in the middle.**

Answer: This sentence has been moved to the beginning of the section and changed to "In this section, we show the impact of atmospheric transport patterns on the atmospheric $CHBr_3$ distribution, with the uniform background $CHBr_3$ emission simulations." in lines 204-205.

**Some specific points here that may or may not be helpful:**
**Line 191: I believe the authors mean "boundary layer mixing ratios" or "mixing ratios in the lower atmosphere" rather than "atmospheric" here?**

Answer: We've replaced "atmospheric $CHBr_3$ mixing ratios" here by "$CHBr_3$ mixing ratios in the lower atmosphere".

**Lines 215 and 220: I think I know what the authors are trying to say, but it's not possible to accumulate the amount of a gas through transport and mixing, and these statements come across as though that's what they are saying. I think they want to state this more carefully, perhaps invoking how wind engages with the ocean surface while diluting what is emitted? Or is it simply about how the winds and mixing and loss redistribute CHBr3?**

Answer: Here we refer to the redistribution of $CHBr_3$ by transport. To avoid ambiguity, we replaced "accumulation" with "higher background mixing ratios". The mixing ratios of the depend on the particles that "fall" within the output grid cells. In the regions of convergence or low wind speeds, where particles could gather or move slowly, it is possible to "accumulate" enough particles to produce high mixing ratios, the description of mixing ratio calculation is added in Sec. 2. In the manuscript, the added part is:

"Output data in form of CHBr$_3$ volume mixing ratios (*VMR*) available at a user-defined grid, were calculated by:

$$VMR = \left(\frac{c_T}{\rho_a}\right) \cdot \left(\frac{m_a}{m_T}\right) \qquad (3)$$

where $c_T$ is the CHBr$_3$ mass concentration, $\rho_a$ is the density of the air, and $m_a$ and $m_T$ are the molecular weight of air and CHBr$_3$, respectively.

For each grid cell, the CHBr$_3$ mass concentration is given by:

$$C_T = \frac{1}{V}\sum_{i=1}^{N} m_i f_i \qquad (4)$$

with $m_i$ being the mass of CHBr$_3$ for particle $i$, $f_i$ the mass fraction of CHBr$_3$ of particle $i$ attributed to the respective grid cell, $N$ the total number of the particles, and $V$ the volume of the grid cell (Stohl et al., 2005). We run FLEXPART in the non-domain filling mode, therefore the particle distribution is not correlated with air density. Particles, and thus bromoform, can accumulate in regions of low wind speeds where the relatively long residence time allows that oceanic emissions constantly add new particles. Similarly, particles can accumulate in regions of convergence where horizontal inflow pools marine boundary layer air from different regions." in lines 185-198.

**Line 222: It's really not a big leap to hypothesize that winds reduce the mixing ratios in the boundary layer. I think the valuable story is about how the wind shear and degradation can influence the distributions.**

Answer: Thanks for the suggestion. We revised this sentence as "the interplay between wind speed and convergence may influence the CHBr3 distribution" and added a new figure (new Fig. 5) to demonstrate it.

In the manuscript, the added sentences are "In order to validate the hypothesis, we show a violin plot of regional background CHBr$_3$ mixing ratios related to convergence/divergence, and to the wind speeds averaged over each simulation period in Fig. 5. For the TransBrom case, the averaged ranges of mixing ratios in regions of convergence and divergence (Fig. 5a) go up to 0.7 ppt and 0.5 ppt, respectively, with interquartile ranges of 0.1-0.35 ppt and 0.05-0.21 ppt. Probability of mixing ratios larger than 0.2 ppt is much higher for regions of convergence compared to regions of divergence. Meanwhile, in the regions grouped by wind speed (Fig. 5b), higher CHBr$_3$ mixing ratios are more likely to occur in regions with lower wind speeds (i.e. in the regions of 0.0-5.0 m/s, mixing ratio go up to 0.65 ppt, while in the regions of 10-15 m/s, mixing ratio go up to 0.25 ppt). Similar distributions also occur for the SHIVA case. During the OASIS case, the CHBr$_3$ mixing ratios are much smaller than for the other two cases due to stronger winds. Highest mixing ratios (~0.15 to ~0.2 ppt) are found in the regions of convergence (Fig. 5e). However, higher mixing ratios are also generally found in the regions of higher wind speeds (Fig. 5f), as the regions of convergence locate in the regions of high wind speed during the OASIS case. The distributions suggest that in general higher CHBr$_3$ mixing ratios tend to occur in the regions of convergence or lower wind speed, with the exception of the OASIS case where extremely high winds occurred and coincided with regions of convergence. " in lines 239-253.

**Lines 224-225: "time period" is redundant; "respectively" is almost always unnecessary.**

Answer: "time period" and "respectively" are deleted.

**Lines 228-229: Again, are these actually accumulations or higher concentrations as the result of dispersal? The former requires emissions of some sort and I don't think that is what this section is about (i.e, constant emissions everywhere).**

Answer: "accumulations" here refers to higher concentrations. See the detailed response to the last comment on this point.

**Atmospheric CHBr3 (hotspots)**
**Observed**
**Lines 289-90. Delete "on the other hand"**
Answer: Deleted.
**Line 290: What is meant by "background" here? I would not use "background' in this paper in any sense other than defined in lines 191-92.**
Answer: The "background" here refers to lower emissions during the campaigns. Sorry for the confusion it caused with the "background emissions" defined previously. We have revised this sentence as "All three campaigns show periods with localized elevated and hotspot emissions.".

**Lines 297-298: How does windspeed dominate the flux? Do the authors mean it is dominating the variability in the flux?**
Answer: "air-sea flux" is replaced by "variability of air-sea flux".

**Lines 306-308: What happened to transport? It's not all just source and loss. And I'm curious, how is loss being handled? I presume it is variable in the atmosphere based on absorption bands etc.?**
Answer: Transport is also included in "loss" here. To avoid ambiguity, we revised the sentence as "interplay of source, transport, and loss processes".
In the manuscript, we described the loss process by adding:
"… Chemical decay of $CHBr_3$ was accounted for by:
$$m(t + \Delta t) = m(t) \exp(-\Delta t/\beta) \qquad (2)$$
where $m$ is the particle mass, $\beta = T_{1/2}/ln(2)$ is the $e$-folding lifetime of $CHBr_3$, and $T_{1/2}$ is the half-life of $CHBr_3$ (Stohl et al., 2005). In our study, a half-life of 17 days ($e$-folding lifetime of 24 days) is prescribed to $CHBr_3$ during all runs (Montzka and Reimann, 2011)." in lines 151-156.

**Lines 339-342: Figure 7 is observations, not models. What is meant here?**
Answer: In the following section we'll compare the atmospheric mixing ratios due to background emissions and localized elevated emissions. We show the figure here because the localized elevated emissions used in the simulations are derived from the observations.

**Lines 349-350: Delete the "hand" phrases. Line 357: "Hand" again. Delete.**
Answer: Deleted.

**Lines 381-384: This is an important statement. Don't lose it during the revision.**

**Summary**
**Just a thought. The summary could be trimmed and used as the abstract of the paper, as it focuses on "what we did, what we found, and what it means", which is all that is really needed in an abstract.**
**Line 392: Replace "revealing" with "demonstrating the role of"; delete "important".**
Answer: Revised as suggested

**Line 401: Insert comma after "location".**
Answer: Inserted

**Line 416: Replace "produce" with "allow".**
Answer: Replaced.

Thanks again for all your valuable comments, hope all your concerns have been addressed.

---

## Author Comment (AC2) · 29 May 2019

Dear Referee #2

We thank the referee for his/her valuable comments, which helped to improve the manuscript. We have added a more detailed descriptions of the model and comparisons with other observational data as well as other model studies. The changed parts in the manuscript are marked in red. The following is a point-by-point response.

**Anonymous Referee #2**

**The authors present an interesting model study on the spatial variation of atmospheric CHBr3 concentrations. Overall the paper is clearly written, and easy to read. While I agree their main conclusion that the spatial distribution is a result of the interplay of transport and emissions (and chemical depletion), I think a major revision is needed before it can be accepted for publication. My major concerns are:**

**1. There is no sufficient description on their transport model. The FLEXPART model has been widely used in various researches. But for CHBr3 simulation, the reader still wants to know the important details, including its description on the air-sea exchange, and on the oxidation scheme etc.**

Answer: More details of the chemical decay (loss process), air-sea flux, and calculations of mixing ratios have been added in section 2. We revised the manuscript by adding:
"The air-sea flux was obtained from the transfer coefficient ($k_w$) and the concentration gradient ($\Delta c$) between water concentration and the theoretical equilibrium water concentration (see details in Fiehn et al., 2017 and reference therein):

$$F = k_w \cdot \Delta c \qquad\qquad (1)"$$

in lines 118-124.

"… Trajectories released from the global ocean surface or along the cruise track carry the amount of CHBr3 prescribed by the respective emission scenario. Chemical decay of CHBr3 was accounted for by:

$$m(t + \Delta t) = m(t) \exp\left(-\Delta t / \beta\right) \qquad\qquad (2)$$

where $m$ is the particle mass, $\beta = T_{1/2}/ln(2)$ is the $e$-folding lifetime of CHBr3, and $T_{1/2}$ is the half-life of CHBr3 (Stohl et al., 2005). In our study, a half-life of 17 days ($e$-folding lifetime of 24 days) is prescribed to CHBr3 during all runs (Montzka and Reimann, 2011)." in lines 150-156.

"Output data in form of CHBr3 volume mixing ratios (VMR) available at a user-defined grid, were calculated by:

$$VMR = \left(\frac{c_T}{\rho_a}\right) \cdot \left(\frac{m_a}{m_T}\right) \qquad\qquad (3)$$

where $C_T$ is the CHBr$_3$ mass concentration, $\rho_a$ is the density of the air, and ma and m$_T$ are the molecular weight of air and CHBr$_3$, respectively.

For each grid cell, the CHBr$_3$ mass concentration is given by:

$$C_T = \frac{1}{V}\sum_{i=1}^{N} m_i f_i \qquad (4)$$

with mi being the mass of CHBr3 for particle i, fi the mass fraction of CHBr3 of particle i attributed to the respective grid cell, N the total number of the particles, and V the volume of the grid cell (Stohl et al., 2005). We run FLEXPART in the non-domain filling mode, therefore the particle distribution is not correlated with air density. Particles, and thus bromoform, can accumulate in regions of low wind speeds where the relatively long residence time allows that oceanic emissions constantly add new particles. Similarly, particles can accumulate in regions of convergence where horizontal inflow pools marine boundary layer air from different regions..” in lines 185-198.

**2. There is no comparison of their simulations with other models, or more importantly with observations. There are questions on how realistic their simulations are. For example, the model shows large areas with very low CHBr3 concentrations (<0.1 pptv), particularly at the 5km level (Figs.8 to 10). They appear inconsistent with aircraft measurements such as the recent CAST and CONTRAST campaigns.**

Answer: The simulations in our paper based on the uniform background emissions, only include open ocean CHBr$_3$ emissions and do not take coastal emissions into consideration. As coastal and shelf emissions contribute significantly to atmospheric CHBr$_3$ mixing ratios, we do not expect our runs to give realistic mixing ratios. The main point of our simulation is not to determine the overall amount of atmospheric CHBr$_3$, but to test whether the signals of localized elevated CHBr$_3$ emissions (hot spots) could be distinguished from those of background emission. To clarify the main point and avoid confusion, we rearranged and rewrote some sections of the manuscript. We added a paragraph in the discussion section to compare our simulations with some aircraft campaigns and model studies explaining differences between the two based on our approach. Several new citations (Harris et al., 2016; Pan et al., 2016; Butler et al., 2018) are also included.

 “Our approach requires that we isolate uniform background CHBr3 emission from coastal and shelf emissions, which can be significant (Fuhlbrügge et al., 2016; Fiehn et al., 2017) and would lead to higher atmospheric abundances. In consequence, we expect the background CHBr3 mixing ratios inferred from our simulations to be smaller compared to observations and other modeling studies. In the Western Pacific (TransBrom), our simulated background mixing ratios at 5 km range from 0.0-0.4 ppt (Fig. 9-11). Measurements from aircraft campaigns in this region, CAST (Harris et al., 2016) and CONTRAST (Pan et al., 2016), show higher CHBr3 mixing ratio of 0.03-0.79 ppt and 0.20-1.127 ppt between 4-6 km. Other model studies (e.g. Hossaini et al., 2016; Butler et al., 2018) based on CHBr3 emission scenarios that include coastal and open ocean sources (e.g. Liang et al., 2010; Ordóñez et al., 2012; Ziska et al., 2013) also suggest the average CHBr3 mixing ratio over 0.5 ppt in this region. ” in lines 423-432.

**3. More detailed analysis is needed. At some places, the paper tends to establish the correlation between the CHBr3 distribution and the (snapshot of) wind fields. Considering the lifetime of CHBr3, such correlations are often not obvious, as**

**demonstration by their own results. Tagged simulations (for example, Butler et al., 2018) may help the reader understand the complexity, particularly for the emission 'hotspots'.**

Answer: To show the correlations in a clearer way, we added a statistical analysis of the CHBr3 distribution by regions of convergence/divergence and wind speeds. The results suggest that higher mixing ratios tend to appear in the regions of convergence or lower wind speed. The statistical results are shown in new Figure 5.

The manuscript is revised by adding "In order to validate the hypothesis, we show a violin plot of regional background CHBr3 mixing ratios related to convergence/divergence, and to the wind speeds averaged over each simulation period in Fig. 5. For the TransBrom case, the averaged ranges of mixing ratios in regions of convergence and divergence (Fig. 5a) go up to 0.7 ppt and 0.5 ppt, respectively, with interquartile ranges of 0.1-0.35 ppt and 0.05-0.21 ppt. Probability of mixing ratios larger than 0.2 ppt is much higher for regions of convergence compared to regions of divergence. Meanwhile, in the regions grouped by wind speed (Fig. 5b), higher CHBr3 mixing ratios are more likely to occur in regions with lower wind speeds (i.e. in the regions of 0.0-5.0 m/s, mixing ratio go up to 0.65 ppt, while in the regions of 10-15 m/s, mixing ratio go up to 0.25 ppt). Similar distributions also occur for the SHIVA case. During the OASIS case, the CHBr3 mixing ratios are much smaller than for the other two cases due to stronger winds. Highest mixing ratios (~0.15 to ~0.2 ppt) are found in the regions of convergence (Fig. 5e). However, higher mixing ratios are also generally found in the regions of higher wind speeds (Fig. 5f), as the regions of convergence locate in the regions of high wind speed during the OASIS case. The distributions suggest that in general higher CHBr3 mixing ratios tend to occur in the regions of convergence or lower wind speed, with the exception of the OASIS case where extremely high winds occurred and coincided with regions of convergence." on lines 239-253.

Thank you again for all your comments, hope the revised manuscript has addressed your concerns.